# Fallback-Enabled Closed-Set Classification: Cross-Modal Consistency in Vision-Language Models

**Sijia Wang**  *sijia.wang@duke.edu*
*Department of Electrical and Computer Engineering*
*Duke University*

**Ricardo Henao**  *ricardo.henao@duke.edu*
*Department of Electrical and Computer Engineering*
*Duke University*

**Reviewed on OpenReview:** *https://openreview.net/forum?id=tOKG6sSk3I*

## Abstract

Vision-Language Models (VLMs) can describe and label images; however, this does not imply that they truly process what they are perceiving. Recent studies show that, despite their breadth of training, VLMs are surprisingly unreliable as classifiers, for either closed-world or open-world settings. In this work, we explore a deeper question: can a VLM recognize when an image falls outside the set of categories it is asked to choose from? Our results reveal a surprising failure mode: even when the notion of in-set versus out-of-set is explicitly defined, VLM models often assign plausible in-set labels to out-of-set images, violating the task's explicit constraint. Motivated by this, we propose a cross-modal consistency framework that reasons over both the visual and textual arms of the model and accepts an answer only when they agree. Experiments on three well-known datasets (DomainNet, VisDA and INaturalist-2021) demonstrate that this approach consistently improves balanced known *vs.* unknown detection over Source-Free Universal Domain Adaptation (SF-UniDA) baselines, showing that cross-modal consistency improves a VLM's ability to follow the task's logic and distinguish when an image falls outside the intended label space. Our results suggest that with strong VLMs, fallback behavior need not rely exclusively on specialized SF-UniDA adaptation pipelines: a lightweight cross-modal consistency decision rule can be competitive with representative SF-UniDA baselines on standard benchmarks.

## 1 Introduction

A long-standing challenge in (visual) recognition is to design systems that not only assign labels among known categories but also detect when an input does not belong to any of the known categories. This challenge has been studied under various names, such as open-world classification (Ding & Pang, 2024; Shu et al., 2017; Fei & Liu, 2016), universal domain adaptation (UniDA) (You et al., 2019b; Chang et al., 2022; Saito & Saenko, 2021; Choe et al., 2025). These formulations differ in scope: open-world classification typically assumes access to labeled training data and learns an unknown detector during training, while UniDA additionally addresses domain shift and partial label overlap. Source-free universal domain adaptation (SF-UniDA) (You et al., 2019a; Liang et al., 2021) further removes the dependency on source data, making adaptation feasible under privacy and efficiency constraints. This is so because it considers the case where one only has access to a pretrained model without access to source data and must adapt the model to a target domain.

The problem studied in these approaches has conceptual *and* practical implications. Epistemically, a system that "knows when it does not know" embodies a basic form of self-awareness long valued in philosophy, as Socrates taught, "wisdom begins by acknowledging the limits of one's knowledge". Practically, in safety-critical cases, *e.g.*, if a medical condition does not belong to the known set of conditions in the diagnostic label set, the medical imaging recognition system should be able to detect it to prevent risky diagnoses.

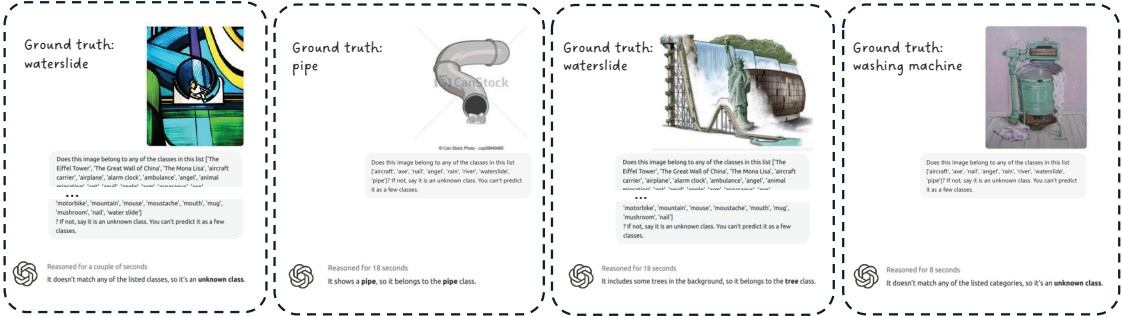

Figure 1: Illustration of the problem. From left to right, we showcase four scenarios, *i*) a known object gets wrongly recognized as unknown, *ii*) a known object gets recognized as known, *iii*) an unknown sample gets wrongly classified as one of the classes in the list because of the background, and lastly, *iv*) an unknown image gets successfully flagged as unknown. Over-rejection error (*i*), which unnecessarily triggers fallback will hurt system efficiency, while overconfidence error (*iii*), bypasses the fallback mechanism, risking unsafe downstream decisions.

The rapid development of Vision-Language Models (VLMs) (Bai et al., 2023; Achiam et al., 2023; Wei et al., 2022) has opened new possibilities to address this challenge (Yin et al., 2023; Kapoor et al., 2024). However, large foundation models suffer from serious hallucination issues Kalai et al. (2025); Huang et al. (2025); Xu et al. (2024). In fact, works such as Chen et al. (2023) found that large foundation models can produce correct answers, but lack internal self-awareness of their own output. Moreover, other works (Zhang et al., 2024; Mitra et al., 2025; Jiang et al., 2025) concluded that VLMs are *bad* at both open-world and closed-world image classification, with training data being the primary cause. Therefore, in order to deploy VLMs to meet the demands of a solution without relying on additional training, alternative procedures need to be taken to make their answers more reliable.

We first explore VLMs on a strict instruction-following setting as illustrated in Figure 1. Given an image and a closed-set label list, the model must assign a label from such a list, and either *i*) declare the image as "known", or *ii*) trigger a fallback decision of "unknown" when the image is judged to fall outside the intended label space. We define this problem as *fallback-enabled closed-set classification* with VLMs. Despite clear instructions and examples, our experiments on the DomainNet dataset expose a paradox: strong categorical labeling ability but weak logical boundary keeping, *i.e.*, very high true-negative rates for unknowns, yet poor true-positive rates for the known set, which results in many in-set images being mislabeled as "unknown". Importantly, clear instructions alone do not make the concept of "unknown" operational or sufficiently clear for these models.

Therefore, in order to use VLMs to address this problem, we propose a framework that tasks the VLM with reasoning along two arms and accepts a prediction only when they agree. Specifically, we make the following contributions.

- We propose a simple, model-agnostic framework that uses cross-modal consistency to perform closed-set classification with a fallback action. The method requires no fine-tuning or access to model internals. Moreover, it is easy to implement with existing VLM APIs.

- We define fallback-enabled closed-set classification with VLMs as a problem that is conceptually aligned with SF-UniDA. We explicitly bridge VLM reasoning with SF-UniDA, analyze their similarities and differences.

- We evaluate the performance of the proposed framework in comparison to the state-of-the-art SF-UniDA methods, not only using harmonic-mean accuracy and overall accuracy, but also the ratio of known *vs.* unknown accuracies as *balance* metric. This ratio reveals when a model is biased toward either over-accepting unknowns or over-rejecting knowns, which is critical for deploying fallback behavior in practice.

- Across six scenarios with three well-known datasets, we show that VLMs equipped with the proposed method can consistently outperform specialized SF-UniDA methods.

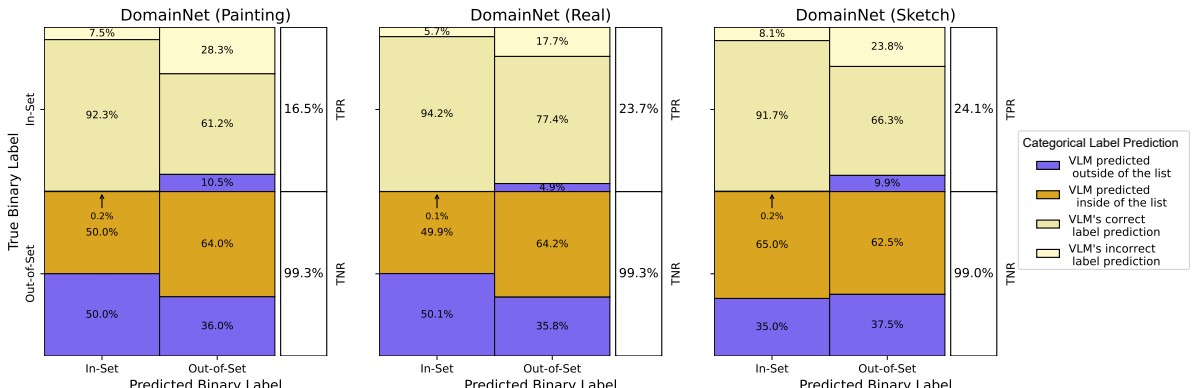

Figure 2: Preliminary analysis demonstrating the paradoxical behavior of VLMs on the DomainNet dataset. Even though the prompt explicitly defines the notions of in-set (known) and out-of-set (unknown) categories, the model fails to internalize this distinction, *i.e.*, it recognizes categorical labels accurately but struggles with binary discrimination between known and unknown samples. High TNR but low TPR indicates that while the model reliably rejects truly out-of-set images, it frequently misclassifies in-set ones as unknown, which reveals that clear instruction alone is insufficient for logical understanding of the concept "unknown".

## 2 A Paradoxical Finding in VLM Probing

We begin by investigating whether VLMs can accurately distinguish between known and unknown categories given a predefined list of labels. Specifically, we ask: *(i)* (Closed-set Classification) Can a VLM accurately determine whether an image belongs to a given list of labels? *(ii)* (Fallback Behavior) Can a VLM recognize when an image lies outside this list and thus should be classified as "unknown"?

We first evaluate the above using a direct prompting approach using GPT 4o-mini on three domains, specifically, Painting, Real, and Sketch, of DomainNet. The prompt we use is as follows, with {this image} and {known class list} being placeholders for the image query and the predefined list of class labels (the details are shown in Appendix A.2).

```
<System Prompt> You are an AI that classifies images based on a predefined list of categories.  If the image belongs to
a category in the GIVEN list (ONLY from the GIVEN list), then provide classname with the correct category name from
the given list and respond with unknown:  False; if the image does not belong to any category in the GIVEN list, then
select the closest possible match from the GIVEN list (DO NOT reply with labels outside of the list) as classname and
respond with unknown:  True.

<User Prompt> Does {this image} belong to one of the categories in the following list {known class list}?  Format the
answer in csv format with keys unknown and classname separated by ','
Example 1:
Image:  (picture of a aeroplane)
Response:  unknown:  False, classname:  'aeroplane'
Example 2:
Image:  (picture of a donkey)
Response:  unknown:  True, classname:  'horse'
```

This prompt explicitly articulates what constitutes an unknown image, encodes the task logic in a system prompt, clearly marks the predefined label set as a given list of classes to make these constraints less likely to be overlooked, and provides concrete formatting examples. These choices, *i.e.*, providing examples and setting the expected output format, follow structured prompting practices for multimodal LLMs to improve adherence to instructions (Sahoo et al., 2024; Zhang et al., 2023; Bsharat et al., 2023).

Figure 2 summarizes the empirical results of two prediction outputs from the VLM (GPT 4o-mini), the categorical label and the binary label "unknown", together in a confusion matrix, for all three domains (Painting, Real, and Sketch) of DomainNet. For each confusion matrix, the rows indicate the ground truth binary label: in-set (known) and out-of-set (unknown), while the columns indicate model predictions. We treat "known" (in-set) as the positive class, so the cells are true positives (TPs) (top-left), false negatives (FNs) (top-right), false positives (FPs) (bottom-left) and false negatives (FNs) (bottom-right). The per-

centage values in each cell denote the prediction accuracies of categorical labels within that subset. With the above definition, $\text{TPR} = \frac{\text{TP}}{\text{TP+FN}}$ measures how many known samples are correctly identified by a model, while $\text{TNR} = \frac{\text{TN}}{\text{TN+FP}}$ shows how often unknown images are correctly recognized. Our observations of the results in this setting revealed significant insights described below.

*Weak known/unknown discrimination.* When we analyze the predicted value of "unknown", the VLM is highly effective at identifying unknown images, correctly classifying these cases as "unknown" with up to 90% accuracy. However, the VLM shows limitations in identifying known images correctly, achieving accuracy as low as 20% in classifying known images as "known". In order to argue that a model is good at detecting what is "unknown", it should be able to have not only a good classification accuracy for "unknown", but also for "known" images. However, the high TNR but low TPR scores in Figure 2 indicate an imbalanced performance to correctly recognize "known" *vs.* "unknown", which does not match the desired behavior.

*Strong label recognition.* Despite poor known-sample detection, the high percentages in the top two cells of each of the confusion matrices in Figure 2 indicate that the VLM achieves high accuracy when assigning labels to known images, thus excelling at closed-set classification.

*Paradoxical results.* These findings suggest that the VLM has a solid label recognition ability (closed-set classification), but struggles with the discrimination of "known" *vs.* "unknown", *i.e.*, the desired fallback behavior, even with explicit instructions for them in the prompt.

*Understanding the paradox.* The results above suggest that the VLM's categorical reasoning and its binary known/unknown judgment are decoupled: the model can correctly identify what an object is, but cannot reliably determine whether that object belongs to the given label list. We hypothesize this occurs because the VLM's pretraining objective focuses on producing descriptive, plausible outputs rather than promoting logical constraint following. A more detailed analysis can be found in Appendix A.7.

## 3 A Solution using Cross-Modal Consistency

Section 2 illustrated that explicit instructions alone cannot reliably realize the fallback mechanism in closed-set classification that we seek to address. Specifically, VLMs may correctly find the category of the known labels, *i.e., the closed set*, but still misjudge in- *vs.* out-of-set membership, *i.e.*, the fallback behavior, even with strict task logic enforced in the prompt wording. A common practice for solving such a problem is to train a classifier with an additional class, thus extending the closed-set with an "unknown" class (Zhan et al., 2021; Shu et al., 2021). Other lines of work, such as UniDA (Liang et al., 2021) and SF-UniDA handle unknowns by thresholding uncertainty scores or synthesizing unknown samples (Bai et al., 2022) in open-world classification. Although effective when source data and retraining are available, these approaches are brittle under domain shift, sensitive to thresholding, and scale poorly as label lists change, especially in source-free settings. Here, we take a different route, by proposing a solution that leverages VLM's broad knowledge without additional training. Before providing the details, we first define the problem studied.

### 3.1 Problem Definition

To define what is "unknown", we first need to recognize what is "known". Given an image $I \in \mathcal{I}$, and a predefined and known list of labels (the closed set) $\mathcal{C} = \{c_1, c_2, \cdots, c_N\}$, with $N$ being the total number of labels in the list, each image has two labels, one is its ground truth categorical label $y$ which may or may not be in $\mathcal{C}$, and the other is a binary label $\delta$ denoting "known" *vs.* "unknown". Specifically, an image is defined as "known" if and only if its ground truth label is within the predefined known class list $C$. We define $\delta$ as

$$\delta = \begin{cases} 1 & \text{if } y \in \mathcal{C} \text{ (known)} \\ 0 & \text{if } y \notin \mathcal{C} \text{ (unknown)} \end{cases} .$$

We seek to understand whether a VLM model $f_{VLM}(\cdot)$ is able to determine whether the image $I$ is "known" and can be appropriately labeled as any label in the set $\mathcal{C}$, or "unknown" and cannot be assigned any label in $\mathcal{C}$. To this end, we propose an approach that utilizes the cross-modal consistency of the VLM for unknown image identification. Specifically, query the VLM along two arms: $i$) prompt directly with the image; and

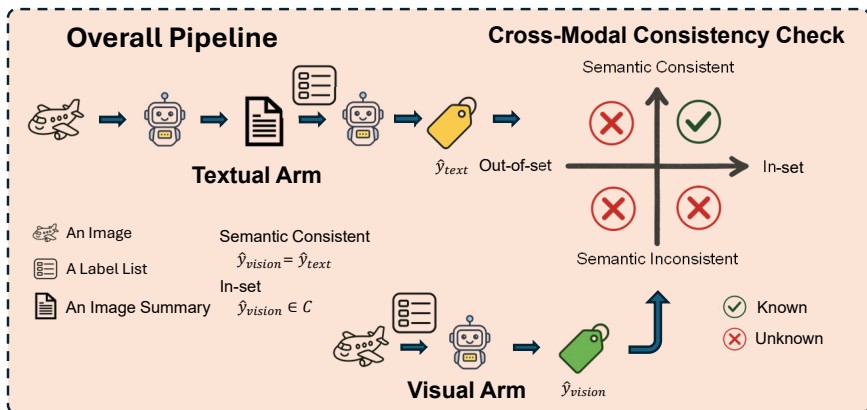

Figure 3: Illustration of the proposed method. The model makes predictions through a visual and a textual arm. A sample is marked known only when predictions from both arms agree on a label within the predefined set; otherwise, it is flagged as unknown.

*ii*) first summarize the image into text, then prompt the same question with such a summary. The details of the proposed framework are described below.

## 3.2 Direct Prompting (Visual Arm)

Given an image $I$, we first directly task the VLM model through prompting with giving a predicted label given the predefined list of known class labels $\mathcal{C}$. We show the partial prompt below for brevity, and the full version is provided in the Appendix A.2.

```
"...  Does this image belong to one of the categories in the following list {known class list}?  Format the answer in csv
format with keys unknown and classname separated by ','..."
```

Let $f_{VLM} : \mathcal{I} \times \mathcal{P} \to \mathcal{R}$, where $\mathcal{P}$ represents the prompt space and $\mathcal{R}$ is the unprocessed response space. We write

$$r_{vision} = f_{VLM}(I, p_{vision}(\mathcal{C})), \tag{1}$$

with $p_{vision}(\cdot)$ being a template for the prompt, which can be found in Appendix A.2.

We then define a parsing function $\phi(\cdot)$ (the detailed definition of which can be found in Appendix A.4) that extracts the categorical class label from the unprocessed response. Specifically, the two possible outcomes are $\hat{y}_{vision} = \phi(r_{vision}) \in \mathcal{C}$ if the parsing is successful and returns a categorical label from $\mathcal{C}$, while $\hat{y}_{vision} = \phi(r_{vision}) = \emptyset$, *i.e.*, the empty set, if the parsing fails or no valid categorical label is found because of failures following the instructions.

This process of generating $\hat{y}_{vision}$ constitutes the model's direct visual reasoning arm, which is based solely on its visual understanding and the label list $\mathcal{C}$.

## 3.3 Summary-Based Prompting (Textual Arm)

Then we task the VLM model to give a summary of the visual features of the image $I$ with the following prompt. We denote the prompt by $p_{text}(\cdot)$.

```
"Can you give a summary of the image?"
```

The generated text $r_{summary} = f_{VLM}(I, p_{text}(\cdot))$ is then used as the context for a second classification prompt shown below (the full version of the prompt can be found in the Appendix A.2).

> "... Does this image belong to one of the categories in the following list {known class list} based on the following summary {image summary}? Format the answer in csv format with keys *unknown* and *classname* separated by ','..."

The output of the VLM given this prompt produces a textual prediction

$$r_{text} = f_{VLM}(r_{summary}).\tag{2}$$

With the same parsing function $\phi(\cdot)$, we can obtain the predicted class label $\hat{y}_{text} = \phi(r_{text})$. Similarly to the direct prompting arm, we expect two outcomes, with $\hat{y}_{text}$ being a parsed string or an empty set.

### 3.4 Cross-modal Consistency of VLMs

Finally, we define the cross-modal consistency criterion to determine whether the model "knows what it knows", thus eliciting the desired fallback behavior. An image is predicted as known only when both reasoning arms agree on the class prediction and such class is within the label list $\mathcal{C}$:

$$\hat{\delta} = \mathbb{1}[\hat{y}_{vision} = \hat{y}_{text}] \cdot \mathbb{1}[\hat{y}_{vision} \in \mathcal{C}],\tag{3}$$

where $\mathbb{1}[\cdot]$ is the indicator function. If the two predictions disagree or the predicted label is not a member of $\mathcal{C}$, the image is labeled "unknown". This dual-arm validation filters out internally inconsistent predictions, reducing overconfidence from a single modality. Although prior work (Manakul et al., 2023; Wang et al., 2022) has considered multi-arm reasoning for reliability, we focus on the minimal and interpretable two-arm case due to the consideration of computational cost. Figure 3 illustrates the full workflow of the proposed cross-modal consistency approach. Given an image and a closed set of labels $\mathcal{C}$, it leverages the VLM by using two decision arms: a visual arm that takes $(I, \mathcal{C})$ and a textual arm that uses a generated description of that image, each of which produces a categorical label. An image is only accepted as "know" when the predicted labels from both arms agree and is in the closed set $\mathcal{C}$; otherwise, it produces the fallback behavior, thus marking the prediction as "unknown". This procedure does not require training, source data (for the closed set $\mathcal{C}$) and is agnostic to the choice of VLM, thus specifically defining a decision rule for "unknown" labels rather than using separate classifiers (Qu et al., 2023) or thresholds on uncertainty estimation as in SF-UniDA approaches (Qu et al., 2023; Liang et al., 2021).

We also ask why the textual arm compensates for the alignment gap. Despite extensive training to align visual and textual representations, a residual misalignment persists between the two spaces - as evidenced by the paradox in Section 2, where categorical labeling and image summarization (which the alignment was directly optimized for) succeeds, but logical set-membership judgment fails. The textual arm addresses this by converting the image into a text summary before classification, effectively collapsing the cross-modal problem into a uni-modal one. Once the visual content has been rendered as text, the comparison between the image description and the label list occurs entirely within the textual space, where language models are natively strong at logical reasoning. The summary thus acts as an explicit bridge that forces visual information through a linguistic bottleneck, narrowing the alignment gap at the point where the boundary decision is made. Agreement between the two arms then combines complementary strengths: the visual arm contributes direct perceptual credibility, while the textual arm contributes reliable within-modality logical reasoning, and disagreement signals that the cross-modal alignment has likely failed for the given image - precisely the cases where a conservative "unknown" fallback is appropriate.

## 4 Comparison to Source-free Universal Domain Adaptation

The fallback-enabled closed-set classification problem we study shows a close resemblance to Source-Free Universal Domain Adaptation (SF-UniDA) (You et al., 2019a; Liang et al., 2021; Qu et al., 2023; 2024). In SF-UniDA, one only has access to a pretrained source model for $\mathcal{C}$ (its training data are inaccessible), and the objective is to adapt this classification model to a new target domain so that samples from the known classes $\mathcal{C}$ are correctly classified, while unknown categories are reliably marked as "unknown". For example, in GLC (Qu et al., 2023), they leverage adaptive global one-vs-all clustering and local consensus clustering to effectively separate known and unknown samples. LEAD (Qu et al., 2024) proposes a decomposition-based learning framework that disentangles the source space into known and unknown components, thus

improving its robustness over general category-shift scenarios. These works underscore growing efforts to develop approaches that upcycle pre-trained models into new domains without accessing source data.

Large-scale VLMs share a similar trait: their training data is hidden from the user, yet they are expected to perform classification when prompted with a given set of labels. Furthermore, in the problem we study, since both VLMs and SF-UniDA models operate as predictors without access to source data, the nature of the problem is fundamentally similar in the sense that the objective is to elicit a fallback mechanism by distinguishing known versus unknown categories from model outputs.

Nevertheless, there are critical distinctions between these two settings. Specifically, SF-UniDA assumes a specific source-target domain relationship with partially overlapping label spaces (You et al., 2019a), whereas VLMs are foundation models trained on broad and multimodal corpora whose "source domain" is implicit and unobservable. Consequently, while SF-UniDA focuses on adapting decision boundaries between known and unknown target samples, our work analyzes how a large-scale VLM internally reasons about task-defined label boundaries when its pretraining data is inaccessible. These comparisons indicate that our cross-modal consistency framework can serve as an *effective alternative* to adaptation-based SF-UniDA pipelines for implementing fallback with strong VLMs. Later, we will show empirically that our rule matches or surpasses representative SF-UniDA baselines on the evaluated benchmarks in our setting.

*Note on Capacity Difference:* SF-UniDA baselines (GLC, LEAD) use ResNet-50 backbones ($\approx 25M$ parameters) trained on source data, while VLMs used in our experiments range from 7B (Qwen-2.5-7B-VL) to over 40B parameters (GPT-4o-mini, Gemini-2.0-flash). The comparison demonstrates that cross-modal consistency is an effective decision rule for the fallback task, but it should not be interpreted as a capacity-matched evaluation. We include SF-UniDA methods as structural reference points because the underlying task (classify knowns, reject unknowns, no source data access) is the same. To provide a more interpretable capacity-controlled reference, we additionally include open-set recognition with CLIP as a baseline.

A more extensive description of the **Additional Related Work** can be found in Appendix A.1.

## 5 Experiments

**Datasets** We evaluated the proposed method on two well-known domain adaptation benchmarks, **DomainNet** (Peng et al., 2019) and **VisDA** (Peng et al., 2017), to compare against state-of-the-art UniDA approaches. **DomainNet** is a large-scale benchmark that contains 345 classes, with approximately 48K–172K images per domain. Following prior work (Qu et al., 2023; 2024), we conduct experiments with the Painting (P), Real (R), and Sketch (S) domains. **VisDA** is another challenging benchmark consisting of 12 object classes, where the source domain includes synthetic object renderings and the target domain consists of photo-realistic images.

In addition, we construct two datasets derived from **INaturalist-2021** (iNaturalist 2021 competition dataset) based on different taxonomic hierarchies, *phylum* and *class*. We explain the details of how the datasets are constructed in the Appendix A.9. The full dataset contains nearly 2.7M images across 10,000 species, with the phylum taxonomy yielding 13 classes and the class taxonomy yielding 51 classes. We selected **INaturalist-2021** for two main reasons. One is that it allows us to evaluate how well the models perform when the class imbalance issue is more severe, and the other is that it enables us to study how our method behaves when the labels are scientific terms instead of common words. To further illustrate the challenge of class imbalance, we visualize the distribution of sample sizes per class in Figure 5 in Appendix A.9.

**Baselines** *(1)* GLC (Qu et al., 2023): separates known and unknown data through adaptive one-vs-all clustering with Silhouette-based pseudo-labeling, which adaptively handles category shifts without prior knowledge. *(2)* LEAD (Qu et al., 2024): applies orthogonal decomposition to separate features associated with known and unknown source samples, thus allowing instance-level identification in target data. *(3)* Self-Consistency MV (Majority Vote)(Wang et al., 2022): queries the visual arm $k$ times with temperature sampling; the final prediction is accepted only when a majority of the $k = 3$ runs agree on the same label in the predefined list. Otherwise, the image is flagged as "unknown". This baseline is run on open-source models only to control for API cost. *(4)* Self-Consistency w/ R (Rephrasing)(Khan & Fu, 2024): queries the visual arm $k = 3$ times with different rephrasings of the user prompt for visual and textual arms (see

Appendix A.3 for exact wording). *(5)* Open-set recognition with CLIP (ViT-L/14) (Miller et al., 2024): treats open-set recognition as a query-set problem: a VLM compares an image embedding to a finite set of label embeddings and uses negative embeddings/words so that the model can reject an input as unknown instead of forcing a wrong class. In the experiment, we show results with 1000 random words as negative embeddings for a more balanced known *vs.* unknown results.

**VLM Models** We consider four VLMs, including two commercial APIs: GPT 4o-mini (Achiam et al., 2023) and Gemini-2.0-flash (Comanici et al., 2025), and two open-source models: LLaMA-3.2-Vision (Grattafiori et al., 2024) and Qwen-2.5-7B-VL (Bai et al., 2025). The implementation details can be found in Appendix A.5.1. Moreover, for reproducibility purposes, the source code is also included in the supplementary material and will be publicly available upon publication.

## 5.1 Metrics

$H$ **score**: $H = \frac{2 \cdot \overline{acc_k} \cdot acc_u}{\overline{acc_k} + acc_u}$ is the harmonic mean between the average of per-class accuracies ($\overline{acc_k}$) over the $K$ classes in $\mathcal{C}$, and the binary accuracy of the unknown samples ($acc_u$).

*Advantages:* The $\overline{acc_k}$ focuses on maintaining discrimination within known classes. The average per-class accuracy prevents the dominant classes from masking the poor performance of rare known classes, while the binary unknown accuracy $acc_u$ measures the task of unknown detection. When combined by harmonic mean, one intends to capture both aspects (closed-set classification and unknown detection) of the performance of UniDA methods.

*Limitations:* Since $\overline{acc_k}$ is the unweighted mean average accuracy over $K$ classes, class imbalance can distort its value, *i.e.*, the rare classes receive equal weights as the more frequent ones. If these infrequent classes have poor performance, then $\overline{acc_k}$ can be severely penalized by the poor performance of classes with few samples. Furthermore, since the harmonic mean is dominated by the component with the smaller value, the $H$ score primarily reflects the worse metric, thus it cannot show whether the model is better with known or unknown classes. In practice, we often wish to evaluate not only the absolute performance on known and unknown samples separately (*e.g.*, their respective accuracies), but also the balance between them, that is, whether the model performs comparably well in recognizing known categories and rejecting unknown ones.

In order to address the limitations of the $H$ score, we also separately show the **weighted accuracy** for known and unknown classes. The weighted accuracies $acc_{k-w}$ and $acc_{u-w}$ incorporate class sample sizes to better reflect the influence of class imbalance. To further show whether each model has a balanced performance for known *vs.* unknown, we also show the **balance ratio** between weighted known accuracy and weighted unknown accuracy.

$$acc_{k-w} = \frac{\sum_K n_k \cdot acc_k}{\sum_K n_k} \quad (5), \quad acc_{u-w} = \frac{\sum_M n_m \cdot acc_m}{\sum_M n_m} \quad (6) \quad R = \frac{acc_{k-w}}{acc_{u-w}} \quad (7)$$

where $k$ is the number of known classes in $\mathcal{C}$, $M$ is the number of unknown classes (often called *private classes* in UniDA), which are assumed to be known only for evaluation purposes, $acc_k$, which is weighted by sample size $n_k$, is the accuracy of the known class $k$, while $acc_m$, the accuracy of the unknown class $m$, is weighted by sample size $n_m$. We expect a good model to have good and balanced performance for both known and unknown classes. Therefore, the closer the ratio $R$ is to 1, the more balanced the model is. For comparison, we also show the mean accuracy for known classes *vs.* unknown classes in Appendix A.8. Furthermore, in the ablation study, we also evaluate the $H$ score between the mean accuracy of the known classes and the accuracy of each private class, individually, specifically $H_{u-m} = \frac{2 \cdot \overline{acc_k} \cdot acc_{u-m}}{\overline{acc_k} + acc_{u-m}}$, where $acc_{u-m}$ indicates the accuracy of the unknown samples from the unknown class $m$.

# 6 Results

We evaluate the proposed cross-modal consistency framework with the four VLMs across six scenarios in DomainNet, VisDA, and INaturalist-2021, as described above, to assess its ability to distinguish known and unknown categories compared to the state-of-the-art (SoTA) source-free UniDA (SF-UniDA) methods.

Table 1: *H* score (%) comparison with SoTA SF-UniDA methods. The first two rows show the SF-UniDA baseline results, cited from Qu et al. (2024). The last four rows show results with VLM models using our cross-modal consistency method (best in bold and second best underlined).

| Method | DomainNet | | | VisDA | INaturalist | | Avg |
|---|---|---|---|---|---|---|---|
| | Painting | Real | Sketch | | Phylum | Class | |
| GLC | 59.1 | 50.5 | 50.7 | 73.1 | 32.4 | 47.8 | 53.8 |
| LEAD | 52.5 | 62.5 | 51.2 | 76.8 | 50.7 | 46.9 | 56.3 |
| OSR w/ CLIP | 61.0 | 75.1 | 60.8 | 68.7 | 38.9 | 12.0 | 52.8 |
| Self-Consistency MV (Llama): | 58.6 | 59.2 | 53.9 | 33.8 | 25.9 | 43.1 | 45.8 |
| Self-Consistency MV (Qwen): | 51.8 | 56.1 | 52.3 | 34.1 | 68.2 | 44.8 | 51.2 |
| Self-Consistency w/ R (Llama): | 61.1 | 65.8 | 62.9 | 30.1 | 18.9 | 57.5 | 49.4 |
| Self-Consistency w/ R (Qwen): | 51.3 | 56.1 | 50.2 | 45.1 | 44.8 | 5.24 | 42.1 |
| GPT 4o-mini (w/ Ours) | 67.2 | 72.4 | 70.2 | 54.0 | 56.0 | 50.5 | 61.7 |
| Gemini (w/ Ours) | 69.9 | 74.0 | 73.3 | 68.2 | 57.5 | 53.3 | 66.0 |
| Llama (w/ Ours) | 63.0 | 68.6 | 56.8 | 65.0 | 54.3 | 36.1 | 57.3 |
| Qwen (w/ Ours) | **77.3** | **85.7** | **80.1** | **78.2** | **73.1** | **57.5** | **75.3** |

Table 2: Comparison to SoTA SF-UniDA methods on weighted accuracies for known ($acc_{k-w}$) and unknown ($acc_{u-w}$) classes. The results are formatted as $acc_{k-w}/acc_{u-w}$. (best in bold and second best underlined).

| Method | DomainNet | | | VisDA | INaturalist | | Avg |
|---|---|---|---|---|---|---|---|
| | Painting | Real | Sketch | | Phylum | Class | |
| GLC | 45.9/77.3 | 49.1/80.5 | 37.3/77.7 | 69.1/67.5 | 61.7/29.9 | 87.8/76.0 | 56.8/68.1 |
| LEAD | 41.1/68.7 | 51.9/73.3 | 34.8/69.7 | 72.6/87.1 | 77.6/65.6 | 63.5/64.6 | 56.9/71.5 |
| OSR (w/ CLIP) | 60.4/63.0 | 77.3/72.8 | 58.5/62.0 | 52.1/91.6 | 17.5/56.6 | 9.0/48.1 | 45.8/65.7 |
| **Self-Consistency MV (3V):** | | | | | | | |
| Llama | 69.4/54.4 | 71.4/56.4 | 60.3/52.0 | 65.6/50.8 | 75.0/15.4 | 70.8/46.7 | 68.8/46.0 |
| Qwen | 92.1/42.3 | 97.6/43.1 | 95.4/41.2 | 91.0/36.2 | 95.9/54.4 | 81.2/51.9 | 92.7/44.9 |
| **Self-Consistency w/ R (3V):** | | | | | | | |
| Llama | 66.7/60.4 | 67.5/62.9 | 63.1/46.2 | 66.6/46.8 | 96.7/10.6 | 86.1/47.0 | 74.5/45.7 |
| Qwen | **93.7**/36.2 | **98.2**/39.2 | **96.2**/36.4 | 92.0/30.2 | **99.9**/28.9 | **99.0**/2.70 | **96.5**/28.9 |
| **Visual-only (V):** | | | | | | | |
| GPT 4o-mini | 70.9/36.3 | 85.0/37.3 | 87.0/36.9 | **96.8**/1.72 | 93.6/0.26 | 93.6/2.87 | 87.8/19.2 |
| Gemini | 72.2/21.9 | 78.3/35.4 | 74.0/26.6 | 91.7/6.70 | 98.4/0.07 | 74.9/17.5 | 81.6/18.0 |
| Llama | 60.0/69.2 | 63.5/78.7 | 46.9/79.4 | 60.8/53.8 | 93.6/10.8 | 56.5/51.5 | 63.5/57.2 |
| Qwen | 87.6/46.4 | 89.2/48.9 | 79.8/42.1 | 92.7/8.79 | 95.6/20.8 | 90.0/25.9 | 89.1/32.1 |
| **Texual-only (T):** | | | | | | | |
| GPT 4o-mini | 68.2/40.0 | 80.9/42.2 | 73.7/40.0 | 89.5/2.35 | 94.0/0.26 | 93.3/3.98 | 83.3/21.5 |
| Gemini | 65.6/35.2 | 71.3/47.9 | 62.4/37.0 | 85.2/6.70 | 96.9/0.07 | 90.2/18.5 | 78.6/24.2 |
| Llama | 47.9/70.1 | 55.3/79.8 | 40.1/79.9 | 64.6/53.8 | 83.6/64.6 | 57.0/51.5 | 58.1/66.6 |
| Qwen | 73.6/60.2 | 80.6/68.9 | 70.2/56.6 | 80.9/57.2 | 83.4/42.5 | 82.1/25.9 | 78.5/51.9 |
| **Cross-Modal Consistency (V+T)(Ours):** | | | | | | | |
| GPT 4o-mini (w/ Ours) | 73.9/62.1 | 85.0/62.8 | 81.2/62.1 | 94.3/37.7 | 93.8/42.4 | 86.9/49.0 | 85.9/52.7 |
| Gemini (w/ Ours) | 71.7/68.5 | 74.0/73.7 | 72.7/74.8 | 90.7/54.7 | 97.1/43.7 | 87.9/52.1 | 82.3/61.3 |
| Llama (w/ Ours) | 49.7/**88.9** | 55.7/**91.8** | 42.3/**91.3** | 55.7/**77.0** | 85.6/58.0 | 53.4/**94.4** | 57.1/83.6 |
| Qwen (w/ Ours) | 72.4/85.6 | 83.3/87.4 | 76.0/84.0 | 84.2/73.8 | 83.8/**87.3** | 80.3/94.3 | 80.0/**85.4** |
| **Cross-Modal Consistency (with the best prompt design as reference): ∗: prompt var3, †: prompt original** | | | | | | | |
| Qwen (w/ ours) | 72.4/85.6* | 83.3/87.4* | 76.084.0* | 84.2/73.8* | 96.5/99.9† | 80.9/92.3† | 81.9/87.2 |

As shown in Table 1, across all benchmarks, our cross-modal consistency framework makes VLMs competitive and often outperforms the classical SoTA SF-UniDA baselines. Specifically, Qwen 2.5-7B-VL consistently achieves the highest H scores, surpassing classical UniDA approaches by up to 20% to 25%, while Gemini 2.0-flash consistently ranks second. The results demonstrate that VLMs equipped with cross-modal reasoning can rival or outperform UniDA algorithms that are *specialized* for such tasks. Table 2 further illustrates how well the weighted accuracies are for known and unknown samples, while Table 3 shows their ratio $R$. The ideal behavior of a model is $R \approx 1$ with high scores for both $acc_w$ and $acc_w$. Qwen 2.5-7B-VL achieves both the highest accuracy and the most balanced value for $R$. Gemini 2.0-flash also achieves strong absolute accuracies, but its $R$ values shift away from 1 for certain datasets. In comparison, SF-UniDA methods exhibit a decent average balance ratio at times, yet their weighted accuracies are much lower in general. As a result, even though GLC performance looks "balanced" by ranking second in $R$ value, its overall weighted accuracy cannot compete with Qwen 2.5-7B-VL and the other VLMs that are both more accurate and comparably

Table 3: Comparison to SoTA SF-UniDA methods on the balance ratio $R$. $R \to 1$ indicates balanced behavior for known and unknown recognition (best in bold and second best underlined). The complete table can be found in Table 11 in the Appendix.

| Method | DomainNet | | | VisDA | INaturalist | | Avg |
| --- | --- | --- | --- | --- | --- | --- | --- |
| | Painting | Real | Sketch | | Phylum | Class | |
| GLC | 0.599 | 0.614 | 0.481 | **1.025** | 2.063 | 1.023 | 0.833 |
| LEAD | 0.600 | 0.710 | 0.502 | 0.834 | 1.182 | **0.983** | 0.736 |
| GPT 4o-mini (w/ Ours) | 1.190 | 1.354 | 1.306 | 2.501 | 2.212 | 1.775 | 1.732 |
| Gemini (w/ Ours) | **1.046** | **1.004** | **0.971** | 1.660 | 2.224 | 1.687 | 1.432 |
| Llama (w/ Ours) | 0.560 | 0.611 | 0.463 | 0.722 | 1.476 | 0.565 | 0.733 |
| Qwen (w/ Ours) | 0.845 | 0.953 | 0.905 | 1.141 | **0.960** | 0.852 | **0.943** |

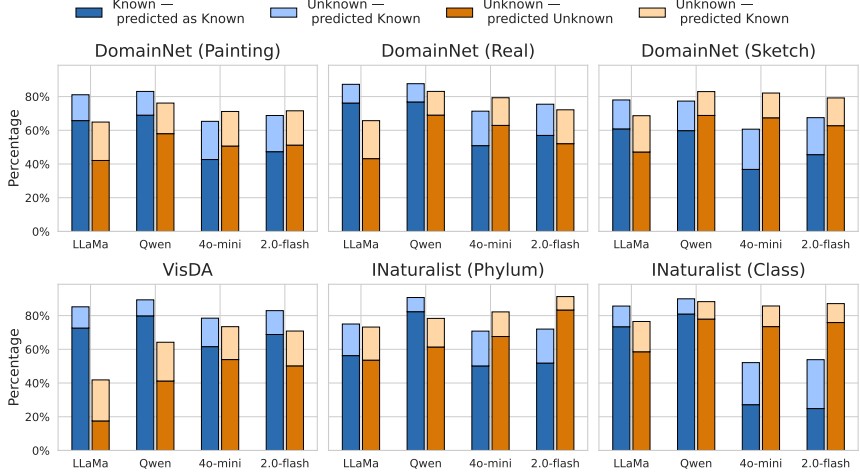

Figure 4: Positive Predictive Value (PPV, blue) and Negative Predictive Value (NPV, orange) of all four VLM models across six datasets. The lighter segments indicate the proportion of samples misclassified within that group for each dataset, while the darker bars correspond to correctly classified known / unknown samples.

(or better) balanced. The two self-consistency baselines consistently achieve a higher known-class accuracy than our cross-modal method, but substantially lower unknown-class accuracy, resulting in lower H scores overall. The cross-modal consistency method primarily improves unknown rejection ($acc_{u-w}$) relative to single-arm baselines, sometimes at a modest cost to known accuracy. This is the expected behavior of an agreement-based decision rule, in which disagreement between the arms makes mistakes on the side of caution by flagging the image as "unknown".

Furthermore, we show positive predictive value (PPV) and negative predictive value (NPV) results in Figure 4 to show how reliable a VLM can be with our framework to behave like it "knows when it knows and when it does not", and observe that Qwen 2.5-7B-VL achieves the most reliable behavior. Overall, these results confirm that enforcing cross-modal consistency framework leads to more stable reasoning and balanced accuracy, reducing the bias between known *vs.* unknown that characterizes classical UniDA baselines.

In the following section, we introduce our ablation studies to demonstrate the influence of prompt variations and removal of framework components.

## 6.1 Ablation Study

**Influence of Prompt Variation** To examine how the image summary influences predictions with our framework, we tested three variations of $p_{text}(\cdot)$: *i*) Variation 1 (var 1) prompts with shorter length of image summaries to evaluate the effect of description verbosity. The motivation of this variation is to test whether reducing linguistic richness helps the models predict more accurately. *ii*) Variation 2 (var 2) "Describe the main object of the image". A prompt that explicitly directed the model to focus on the main object of the image. *iii*) Variation 3 (var 3) "Identify the primary object in the image, excluding any background elements".

Table 4: Ablation Study on prompt variations with Qwen 2.5-7B-VL. We show $H_{u-m}$ scores (%) for the individual target private classes in the VisDA validation dataset. Numbers in each cell represent the results on the validation set.

| | Target Private Class | | | Overall $H$ score |
|---|---|---|---|---|
| | skateboard | train | truck | |
| original | 57.36 | 87.29 | 63.14 | 72.56 |
| var 1 | 58.29 | 87.80 | 61.34 | 72.29 |
| var 2 | 72.75 | **90.56** | 64.00 | 76.91 |
| var 3 | **84.45** | 87.38 | **65.21** | **78.24** |

A prompt that went a step further by specifying that the background should be excluded, restricting attention solely to the primary object. This study allows us to disentangle whether richer contextual descriptions or stronger object-centric guidance lead to more reliable recognition.

As shown in Table 4, both the overall $H$ score and the $H$ score wrt each individual target private class (which is class that does not show in the known label set but only exists in the target dataset) increase for VisDA as the prompts become more object-centric (from original to var 3, the view gets more object-centric). However, when we experiment with a similar ablation study for INaturalist, this trend reverses on INaturalist, for both phylum and class taxonomy, where variation 3 performs worse than the original prompt, the results of which are shown in the Appendix A.8. The explanation for the difference in performance trends lies in the fine-grained taxonomy of the dataset, in which many species can only be distinguished by subtle contextual cues such as texture, color tone, or typical habitat. When the prompt suppresses background information, these discriminative cues are lost and the model's alignment between visual features and (scientific) labels weakens. In addition, scientific names have limited pretraining data of such VLM models, which makes contextual signals even more crucial. As a whole, these results show that the decision of whether to exclude the background is image- and task-dependent. For example, for an image of "an arm with a dreamcatcher tattoo" in DomainNet, the ground-truth label "arm" competes with a plausible interpretation "tattoo", and the "right" choice of label depends on what the user intends to infer. Such image examples are discussed in the Appendix A.10. In summary, these observations highlight that prompt design should respect the semantic granularity of the task: use object-focused prompting enhances logical consistency in structured object domains, whereas prompt with context-aware phrasing for fine-grained classes or ecologically / environmentally dependent recognition.

Importantly, these prompt variations represent explicit, controllable design choices for practitioners and not hidden degrees of freedom that undermine the generality of the proposed framework. By adapting the image summary prompt to match the semantic granularity of the task domain, users can systematically improve cross-modal consistency without requiring retraining or model modifications. This flexibility is a feature, not a limitation, as it enables practitioners to tailor the framework to their specific application context.

Table 5: $H_{u-m}$ scores (%) for the individual target private classes in the VisDA validation set.

| | Target Private Class | | | Overall $H$ score |
|---|---|---|---|---|
| | skateboard | train | truck | |
| GLC | 69.72 | 73.92 | 70.53 | 71.60 |
| LEAD | 75.56 | 57.92 | **76.08** | 77.10 |
| Qwen | **84.45** | **87.38** | 65.21 | **78.24** |

**Discussion of VisDA results** In Table 5, we observe that the VLMs achieve significantly lower $H$ scores compared to the UniDA baselines. To better understand this gap, we examined the experimental setup in detail. The predefined label list contains nine known classes (aeroplane, bicycle, bus, car, horse, knife, motorcycle, person, plant), while the target private set consists of only three classes: skateboard, train, and truck. We calculate $H_{u-m}$ scores for each target private class in VisDA. The results, which are shown in Table 5, reveal that the $H_{u-m}$ score drops substantially when the private class is a truck. A closer inspection of the source data indicates why: the "car" category is represented exclusively by sedans, SUVs, hatchbacks, and sports cars, with no overlap with truck. For methods such as GLC and LEAD, which rely on pretrained source models, this restricted source definition of "car" enables them to more easily separate truck as an unknown class. In contrast, VLMs, trained on broader corpora, likely encode a richer concept of "car"

Table 6: Inference cost and latency analysis.

| Model | Config | Calls | Latency (ms) | Throughput | Cost/100 Img | Total Tokens / Img |
|---|---|---|---|---|---|---|
| **GPT 4o-mini** | V | 1 | $1069 \pm 858$ | 0.935 | $0.057 | 3,748 |
| | T | 2 | $1939 \pm 1098$ | 0.516 | $0.101 | 6,625 |
| | V+T | 3 | $3008 \pm 1455$ | 0.332 | $0.158 | 10,412 |
| **Gemini** | V | 1 | $1222 \pm 505$ | 0.819 | $0.021 | 2,108 |
| | T | 2 | $2329 \pm 220$ | 0.430 | $0.036 | 3,483 |
| | V+T | 3 | $3550 \pm 264$ | 0.282 | $0.057 | 5,591 |

that encompasses truck as a subtype. Consequently, the greater semantic knowledge of VLMs paradoxically makes it harder to avoid misclassifying a truck as a car, leading to a lower $H_{u-m}$ score.

**Influence of Cross-modal Consistency.** To isolate the contribution of our cross-modal consistency mechanism, we ablate three configurations: (i) *visual-only* ($\hat{y}_{vision} \in \mathcal{C}$), (ii) *textual-only* ($\hat{y}_{text} \in \mathcal{C}$), and (iii) our full method combining both arms with consistency. As shown in Table 2, visual-only is prone to a *closed-set bias*: very high known-class weighted accuracy while severely under-rejecting unknowns. For example, on VisDA, Qwen's visual-only variant reaches 92.7/8.79 ($acc_{k-w}$, $acc_{u-w}$), indicating many private-class images are incorrectly accepted as known. Introducing the textual arm helps mitigate this by enabling semantic mismatch reasoning, typically increasing unknown-class accuracy but sometimes at the cost of known accuracy. Our cross-modal consistency framework provides an additional, non-redundant check: by requiring agreement between what the model "sees" and what it can justify from its own summary, it increases recognition of "unknowns" while preserving strong known recognition. The large performance gain shows that by leveraging the consistency between visual and textual reasoning paths, we enhance the model's ability to recognize what it truly knows.

## 6.2 Inference Cost Analysis

For each commercial VLM we report five quantities: *(1)* number of API/model calls per image, *(2)* mean latency per image with standard deviation, *(3)* throughput in seconds per 100 images, *(4)* estimated monetary cost per 100 images, and *(5)* the total token usage per image. We measured latency, throughput, and monetary cost on a 100-image subset on DomainNet in Table 6. We note that the visual and textual arms are independent and can be dispatched in parallel, which would reduce the wall-clock time of V+T to approximately max(V, T) rather than V+T; we leave this engineering optimization to future work.

## 7 Conclusion

We present a cross-modal consistency framework that enables VLMs to better distinguish known from unknown images by requiring agreement between visual and textual classification arms. Experiments on DomainNet, VisDA, and INaturalist-2021 demonstrate that our approach for fallback-enabled closed-set classification consistently improved balanced known *vs.* unknown weighted accuracy and $H$ score relative to SoTA SF-UniDA baselines. The results suggest that enforcing agreement enhances a VLM's logical understanding of "unknown", marking a step toward more self-aware and reliable vision-language reasoning. Moreover, our results suggest that the traditional SF-UniDA paradigm may no longer be the most effective way to obtain fallback behavior when strong VLMs are available. In our experiments, a frozen VLM equipped with a cross-modal consistency framework consistently matches or outperforms SF-UniDA methods, pointing toward a future in which open-world recognition relies more on VLM-based decision rules than on dedicated adapters on pretrained source classifiers.

**Limitations and Future Works** Our study has several limitations. First, we focus on single-label image classification with fallback action, leaving multi-label classification and more complex vision tasks to interesting future work. Second, the proposed framework still requires multiple prompts per sample, which would increase latency and cost in large-scale commercial use. In the future, we plan to explore ways to reduce query cost. Third, because both reasoning arms rely on the same underlying VLM, systematic misunderstandings produce correlated errors that our agreement rule cannot detect. We included an analysis of this type of error in the Appendix A.6. Future work could explore the use of different VLMs for each arm or incorporating calibrated confidence scores to mitigate this.

## 8 Acknowledgments

The authors thank Dr. Dhanajit Brahma for valuable discussions and suggestions during the preparation of this manuscript. This work was supported in part by the Office of Naval Research (ONR) under grant N00014-18-1-2871-P00002-3 and by the National Institutes of Health (NIH) under grant 1R61-NS120246-02.

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
