# OpenReview forum: "Fallback-Enabled Closed-Set Classification: Cross-Modal Consistency in Vision-Language Models"
_TMLR — Accepted by TMLR_

### Review · Reviewer_skZd · 2026-03-13

**Summary Of Contributions:**

The paper observes that VLMs are good at categorical labeling but surprisingly poor at judging whether an image belongs to a given label set. The authors propose a cross-modal consistency check: query the VLM once with the image and once with a generated text summary, and accept a prediction only when both agree. Experiments on DomainNet, VisDA, and INaturalist show improvements over single-pass baselines and competitive results with SF-UniDA methods.

The paradox finding is genuinely interesting. The method is simple and straightforward. The major weakness is that, although the paper includes some ablations (e.g., prompt variations in Section 7.1), it does not sufficiently investigate why VLMs exhibit the paradox observed in Section 2, or what specifically about the consistency check makes it effective beyond general multi-pass agreement.

**Audience:**

Yes

**Audience Explanation:**

The fallback-enabled closed-set classification problem is practically relevant, the Section 2 finding is useful, and the cross-modal consistency framework is lightweight and easy to adopt.

**Claims And Evidence:**

No

**Claims Explanation:**

1. The claim that cross-modal reasoning is the key factor is not sufficiently supported. The paper does not compare against self-consistency baselines that do not involve the cross-modal split, such as sampling from the same arm multiple times with majority vote [1] or rephrasing the prompt and checking agreement [2]. Without these, the gains could come from multi-pass agreement in general rather than the cross-modal design.
2. The claim that the method outperforms SF-UniDA baselines (GLC, LEAD) needs qualification. These baselines use ResNet-scale backbones while the proposed method relies on billion-parameter VLMs.

Reference:

[1] Self-Consistency Improves Chain of Thought Reasoning in Language Models. ICLR 2023

[2] Consistency and Uncertainty: Identifying Unreliable Responses From Black-Box Vision-Language Models for Selective Visual Question Answering. CVPR 2024

**Requested Changes:**

1. Add VLM self-consistency baselines to disentangle the effect of cross-modal design from multi-pass agreement.
2. Include capacity-controlled baselines such as CLIP zero-shot classification under the same consistency rule, to make the SF-UniDA comparison more interpretable.
3. Provide deeper analysis of the Section 2 paradox and why the proposed cross-modal consistency approach works.

---

> ### Author Response · Authors · 2026-04-09
>
> **Requested Changes**
> 1. We fully agree. We **add baselines** in the revision that keep the number of queries comparable to our method while removing the cross-modal split, repeated visual-arm prompting with majority vote [1] for self-consistency, and a rephrase-based [2] baseline. This will allow us to test whether cross-modal agreement contributes beyond simple repetition or prompt rewording.
> 2. We have also added a **capacity-controlled baseline open-set recognition with CLIP**. We also agree that the comparison to GLC/LEAD needs qualification because those methods operate with different backbone scales and assumptions. We also **moderate the statement** that our method “outperforms SF-UniDA baselines" in the revision. A more precise claim is that strong frozen VLMs equipped with a simple consistency rule are competitive with, and often outperform, representative SF-UniDA pipelines on the evaluated benchmarks. This revised wording better reflects the evidence and directly addresses your concern.
> We made this explicit when we compared with SF-UniDA in the revision. We also want to emphasize that our approach **stands as an effective alternative**.
> 3. **On analysis of the paradox**. We agree that this is one of the most interesting aspects of the paper and deserves more analysis. We hypothesize this occurs because the VLM's pretraining objective focuses on producing descriptive, plausible outputs rather than enforcing strict set-membership logic. When asked to simultaneously classify and judge membership, the model defaults to its stronger capability (labeling) at the expense of the weaker one (boundary-keeping). This decoupling motivates the cross-modal consistency approach: by requiring two structurally different reasoning paths to agree, we force the model to "double-check" its boundary decision rather than relying on a single pass where the labeling signal overwhelms the membership judgment. I have **added this in Section 2 in the revision as further explanation. We’ve also added in Section 3.4 in the revision why the proposed cross-modal consistency approach works.**

---

### Review · Reviewer_ea9Y · 2026-03-14

**Summary Of Contributions:**

Summary:

* The paper studies whether Vision-Language Models (VLMs) can reliably perform fallback-enabled closed-set classification, where a model must classify an image into a predefined label set or declare it “unknown” if it does not belong to any listed class.

* Authors show that VLMs exhibit a paradoxical behavior: although they are strong at assigning labels among known categories, they struggle to correctly distinguish between known and unknown samples, often misclassifying in-set images as unknown. Proposed cross-modal consistency framework that queries the VLM through two reasoning paths: a visual arm that directly classifies the image and a textual arm that first generates an image description and then performs classification. A prediction is accepted only when both arms agree on the same label within the predefined set; otherwise, the model outputs “unknown.” The method is carried out on DomainNet, VisDA, and INaturalist-2021 showing some improvements.

Strength:
* The paper is well written, easy to follow, and clearly motivates the problem and proposed solution.

* The proposed framework is straightforward and reproducible with extensive experiments on different VLMs.
Consistent improvements in metrics such as H-score and balanced accuracy.

Weakness:
* The proposed method is completely training-free and the consistency checking across multiple model outputs is conceptually similar to self-consistency reasoning. Thus, the technical novelty is limited.
* The process includes multiple calls: image classification prompt, image summary generation and classification from summary. This results in multiple VLM calls per image, increasing latency and cost for large-scale deployment. However, the authors do not provide detailed reported latency/inference cost in the paper, making it hard to assess the proposed framework.
* Performance depends strongly on: prompt wording and summary quality. Ablation results show that certain prompt variants significantly change performance, suggesting the framework is sensitivity to prompt engineering.
* The framework assumes that the visual arm and textual arm fail independently. In practice, both outputs are produced by the same model, so their errors are often correlated. If the model systematically misunderstands the object, both arms may agree on the same wrong label, meaning the method cannot detect that error.
* The method is essentially a heuristic decision rule (agreement between two prompts). The paper does not provide theoretical analysis of why cross-modal agreement should improve unknown detection, thus, it lacks formal grounding
* The baselines are primarily SF-UniDA methods and the paper does not compare against well-established baselines: modern open-vocabulary detectors [1,2,3], CLIP-based open-set recognition methods [4,5], energy-based or confidence calibration approaches [6]. This makes it unclear how the method performs relative to current open-set recognition techniques.


Reference:

[1]: Cheng, Tianheng, et al. "Yolo-world: Real-time open-vocabulary object detection." Proceedings of the IEEE/CVF conference on computer vision and pattern recognition. 2024.

[2]: Liu, Shilong, et al. "Grounding dino: Marrying dino with grounded pre-training for open-set object detection." European conference on computer vision. Cham: Springer Nature Switzerland, 2024.

[3]: Carion, Nicolas, et al. "Sam 3: Segment anything with concepts." arXiv preprint arXiv:2511.16719 (2025).

[4]: Radford, Alec, et al. "Learning transferable visual models from natural language supervision." International conference on machine learning. PmLR, 2021.

[5]: Alayrac, Jean-Baptiste, et al. "Flamingo: a visual language model for few-shot learning." Advances in neural information processing systems 35 (2022): 23716-23736.

[6]: Joseph, K. J., et al. "Towards open world object detection." Proceedings of the IEEE/CVF conference on computer vision and pattern recognition. 2021.

**Additional Comments:**

N/A

**Audience:**

Yes

**Audience Explanation:**

The paper investigates the reliability of VLMs in distinguishing between known and unknown categories and proposes a simple cross-modal consistency framework to improve fallback behavior without additional training. This framework could flexibly adapt to many VLMs, given its simplicity and training-free design.

**Claims And Evidence:**

Yes

**Claims Explanation:**

The paper presents a clear and simple framework supported by comprehensive quantitative results. Overall, the experimental evidence appears well grounded and consistent with the proposed approach, providing convincing support for the authors’ claims.

**Requested Changes:**

I encourage the authors to address each of the weaknesses outlined above in a point-by-point manner.

---

> ### Author Response · Authors · 2026-04-09
>
> We thank Reviewer ea9Y for the positive assessment of the paper’s clarity, reproducibility, and empirical consistency. We agree that the current draft should better position the novelty and more fully discuss cost, sensitivity, correlation of errors, theoretical intuition, and the choice of baselines.
>
> **W1** - We agree that the method is simple and shares spirit with self-consistency. In the revision, we have **added self-consistency baselines** (majority vote and rephrase) in Section 6 on the open-source models. We also **added a paragraph in Section 3.4 explaining why cross-modal consistency differs**. We believe this sharper positioning is better aligned with what the current evidence supports.
>
> **W2** - New Section 6.2 reports per-image call counts and approximate **latency/cost** for each method configuration (see Table A in the revised paper).
>
> **W3** - We agree that prompt dependence should be made more explicit. Our current ablations already show that object-centric prompting can help on structured object datasets like VisDA but may disadvantage fine-grained settings such as iNaturalist, where contextual cues matter. In the revision, we **emphasize that the framework works in general, prompting is a controllable design choice rather than a hidden degree of freedom**, and add clearer guidance on when object-only versus context-aware summaries are appropriate. We also added a short note in the revision in Section 6.1 to stress this.
>
> **W4** - We now explicitly acknowledge this **limitation in Section 7** (Limitations) and **add an correlation error analysis in the Appendix A.6**. We state that when the model has a **systematic misunderstanding, both arms may agree on the same wrong label**. This is a ceiling of any agreement-based approach using a single model, and we suggest using different VLMs for each arm as future work. At the same time, the current visual-only, textual-only, and full-method **results already show that the second arm is not redundant in practice because it shifts the known/unknown tradeoff substantially**.
>
> **W5** - We agree that a formal theoretical justification is beyond the scope of the current paper, since cross-modal misalignment is a challenging problem in general for VLM studies, but we can **provide a stronger formal intuition**. In the revision Section 3.4, we add a short proposition-level discussion explaining that the rule acts as a conservative acceptance criterion: it predicts “known” only on the intersection of two decision events, which reduces over-acceptance unless both arms share the same error. We present this as an intuition for improved fallback behavior, not as a complete guarantee.
>
> **W6** - We thank the reviewer for suggesting these baselines. We examined each carefully and found that [1–3, 6] target object detection or segmentation (outputting bounding boxes or pixel masks), not image-level classification with fallback. **Adapting them to our setting would require non-trivial modifications** (box-to-label aggregation, confidence thresholds for unknown detection) whose design choices would confound the comparison. [5] (Flamingo) requires in-context examples, changing the problem from zero-shot to few-shot. [6] further requires access to model logits, unavailable for black-box VLM APIs. In contrast, **CLIP zero-shot [4] is directly comparable** -- it performs training-free image-level classification under a fixed label list -- and we have **added it as a capacity-controlled baseline in the revision**. We **discuss the mismatch of [1–3, 5–6] with our setting in the revised Section 5 and Appendix A**.

---

### Review · Reviewer_Rb3M · 2026-03-25

**Summary Of Contributions:**

Proposes a method to use VLM for the classification of images. Specifically, the VLM is given a set of possible classes and needs to declare which class it is or whether it is unknown. The proposed method improves the output of the VLM by cross-checking the consistency of the output from two different pipelines. Once directly from the given image, and once by first generating a summary of the image and then predicting the output only from the text. The performance is compared with source-free universal domain adaptation on three different datasets.

Strengths
- Interesting analysis of the capabilities of VLMs for the chosen task.
- Simple approach on how to use VLMs for the task.
- Tested 4 different VLMs.


Weaknesses
- Although described as a common challenge, the proposed method is only compared to two baselines, specifically source-free universal domain adaptation. However, these methods have a much harder task to solve. VLMs might have seen all the tested domains during training and therefore have a clear advantage over the baselines. Therefore, methods that use training should also be included to compare the predictive performance of the proposed non-training approach. That would give a stronger baseline and give a better impression of how good the proposed approach is. It would be interesting to see whether VLMs can keep up with them.
- The paper is written very lengthy and rather complicated. E.g., the discussion of related work is distributed over multiple sections, the motivation in the introduction is rather complicated formulated, e.g., the choice of baselines at that point is unclear (only gets explained in sec. 4) and inconsistent terminology.
- The proposed cross-modal consistency is mainly beneficial for the unknown classes, while the effect on the known classes is mixed. Unclear how to draw any conclusions from the presented results in Tab. 1. Especially, as the ablation study also shows that the performance can be heavily dependent on the prompt and therefore the prompt needs to be adapted to the task. Therefore, Tab. 1 should additionally include the results with the best prompt design for each dataset to provide a more informative picture.

Minor
- The way the citations are formatted makes the sentences difficult to read. Either the references should be in brackets or really part of the sentences in the form, e.g., Author et al. propose ...
- The references do not link directly to the bibliography, also does not work for references to tables and figures.
- The provided link for the code seems to lead to an almost empty repository, only containing a readme with the paper title.

Unclear usage of terminology
- Why defining the problem as fallback-enabled closed-set classification, how does it differ from the task of open-world classification as described at the beginning of the introduction?
- Open-world classification and universal domain adaptation is more than just detecting when an input does not belong to any of the know categories. The beginning of the introduction is misleading in this regard.
- Sec. 2 lists three questions, however, the third one is the combination of the previous two. Unclear why this is separately listed.



- I would argue, given the instruction, the example (iii) in figure 1 is not completely wrong, as there are indeed trees in the image.

**Audience:**

No

**Audience Explanation:**

Generally, yes, specifically leaning towards no due to how the findings are presented in the paper, the knowledge transfer is limited.

**Broader Impact Concerns:**

no concerns discussed

**Claims And Evidence:**

Yes

**Claims Explanation:**

The specific claims made are supported with evidence. However, the comparison is claimed and done in a very restricted setting, i.e., limited to the comparison of source-free domain adaptation, and therefore the insights gained are limited. The paper would be more interesting if the capabilities of VLMs for the described task were evaluated more broadly, using a wider range of baselines.

**Requested Changes:**

- Better structure the paper, be more concise and precise in writing.
- Include more baselines, also training-based approaches, to provide a clearer picture of the claimed capabilities of VLMs for the task.

---

> ### Author Response · Authors · 2026-04-09
>
> We thank Reviewer Rb3M for the thorough review. We address each concern below.
>
> **W1** - We agree that a **broader baseline comparison** strengthens the paper. In the revision, we have added three new baselines.
>
> 1. **VLM self-consistency baselines**, including visual-arm with **majority vote** and **paraphrasing-based** methods, and to test whether the gain comes merely from multiple queries or specifically from the visual-summary split. To make it a fair comparison (cost-wise), we prompt it three times and get the majority vote out of three.
>
> 2. **Open-set recognition** with CLIP ViT-L/14, where they used negative embeddings to help with unknown detection.
>
> **W2** - We have **restructured the paper** as follows:
>
> 1. **Moved** the full Section 5 (**"Additional Related Work"**) to Appendix A.1.
>
> 2. **Clarified terminology** in the Introduction: We have clarified the differences between open-world classification and universal domain adaptation in the revised Introduction. Open-world classification typically assumes access to labeled training data for known classes, and it learns an unknown detector during training.
>
> **W3** - The idea is that we are **simultaneously improving unknown detection without sacrificing too much of the known class's accuracy**. We agree that **prompt dependence** should be made more explicit. Our current ablations already show that object-centric prompting can help on structured object datasets like VisDA but may disadvantage fine-grained settings such as iNaturalist, where contextual cues matter. In the revision, we emphasize that **the framework works in general, prompting is a controllable design choice rather than a hidden degree of freedom**, and add clearer guidance on when object-only versus context-aware summaries are appropriate. We added a short note in the revision in Section 6.1 Influence of Prompt Variation to stress this.
>
> We also add the **results with the best prompt design** for each dataset in Table 1 as reference.
>
> **Minor** - The citation format is fixed throughout the revision. Internal references to tables, figures, and bibliography have been checked and fixed. And the code repository has been updated with scripts of our codebase.
>
> **Unclear usage of terminology**
> 1. Our setting is strictly inference-time compared to open-world classification: a frozen VLM is given a fixed label list and must either select a label or abstain. We use "fallback-enabled closed-set classification" to emphasize that the label space is closed and given, and the model's job is to follow this constraint.
> 2. We have clarified this in the revised introduction.
> 3. We combined question 2 and 3 accordingly in the revision in Section 2.
> 4. We agree that this is a valid observation, and it actually reinforces our point. The presence of trees in the background makes the 'tree' label semantically plausible, which illustrates exactly why the fallback problem is hard: background cues can mislead the model into assigning an in-set label to an out-of-set image.

---

### Author Response · Authors · 2026-04-09
**Common response**

We sincerely thank all three reviewers for their constructive and detailed feedback. We are encouraged that all reviewers find the paradox finding "genuinely interesting" (Reviewer skZd), the framework "straightforward and reproducible" (Reviewer ea9Y), and the analysis of VLM capabilities "interesting" (Reviewer Rb3M). Below, we address each reviewer's concerns point by point, describing the changes made in the revised manuscript.

We also appreciate the reviewers’ central concern that the previous version does not yet fully disentangle the effect of cross-modal consistency from multi-pass agreement more generally. In the revision, we make this distinction explicit, weaken any wording that implies a stronger causal claim than our current evidence supports, and reposition the contribution as a simple, training-free decision rule for fallback-enabled closed-set classification, together with the empirical finding that this rule is effective across multiple VLMs and datasets, which underscores its utility in practice where training may be prohibitive or ease of deployment may be a priority.

Concretely, we revise the paper mainly in four ways.

**First**, we add **self-consistency baselines**, including repeated visual-arm prompting with majority vote and paraphrase-based agreement within the same arm, to test whether the gain comes merely from multiple queries or specifically from the visual-summary split. To make it a fair comparison (cost-wise), we prompt it three times.

**Second**, in the revision, we explicitly point out **the comparisons to SF-UniDA by** noting that our method uses larger frozen VLMs, while GLC and LEAD rely on smaller task-specific **backbones**.

**Third**, we include an **analysis of the Section 2 paradox**, focusing on the separation between strong categorical recognition and weak in-/out-of-set boundary keeping, which has already been observed in the confusion matrix analysis and the visual-only/textual-only/full ablations.

**Fourth**, we add a short **latency/cost discussion**, reporting the number of model calls per sample and wall-clock/API cost measurements for representative settings.

We also sharpen the presentation of **limitations**.

---

### Decision · Action_Editor_pwRZ · 2026-05-11

**Recommendation:** Accept as is

**Additional Comments:**

It is observed in this paper that even though VLMs are good at labeling among known classes, they often fail at determining whether an image belongs to a given label set. To deal with this problem, a rule based on cross-modal consistency is proposed, where a prediction is accepted only when direct image classification and text-based classification via a generated image summary are in agreement. The paper formalizes this “fallback-enabled closed-set classification” problem and shows that this simple consistency rule is effective across multiple VLMs and datasets.

The final recommendations from the reviewers were split, with two leaning to accept and one leaning to reject. Among the remaining concerns are the usefulness of this method given the lower performance on known classes in exchange for improved unknown detection, dependence of performance on prompt design, and the restrictiveness of the proposed task. These criticisms are all valid but are viewed by the action editor as non-fatal limitations that are acknowledged in the paper and sufficiently offset by the insight-driven contribution and strong empirical validation. It was thus judged that the paper meets the standards for publication in TMLR.

**Audience:**

Yes

**Audience Explanation:**

The paper introduces an interesting problem that arises in VLM-based classification, and the proposed solution, though simple and largely heuristic, is effective and well-validated. The action editor concurs with the reviewers that this work is of interest to TMLR.

**Claims And Evidence:**

Yes

**Claims Explanation:**

The central claim that cross-modal consistency improves VLM-based known/unknown discrimination is now adequately supported after clarifications and certain baselines were added in the revised manuscript. The difference in modeling capacity weakens the SF-UniDA comparisons to some degree, but this issue has been acceptably handled by moderating the claim regarding these methods.